# Observation of unusual outer-sphere mechanism using simple alkenes as nucleophiles in allylation chemistry

Yaxin Zeng[1], Han Gao[2], Zhong-Tao Jiang[1], Yulei Zhu[1], Jinqi Chen[1], Han Zhang[1], Gang Lu ⓘ [2] ✉ & Ying Xia ⓘ [1] ✉

Transition-metal catalyzed allylic substitution reactions of alkenes are among the most efficient methods for synthesizing diene compounds, driven by the inherent preference for an inner-sphere mechanism. Here, we present a demonstration of an outer-sphere mechanism in Rh-catalyzed allylic substitution reaction of simple alkenes using *gem*-difluorinated cyclopropanes as allyl surrogates. This unconventional mechanism offers an opportunity for the fluorine recycling of *gem*-difluorinated cyclopropanes via C − F bond cleavage/ reformation, ultimately delivering allylic carbofluorination products. The developed method tolerates a wide range of simple alkenes, providing access to secondary, tertiary fluorides and *gem*-difluorides with 100% atom economy. DFT calculations reveal that the C − C bond formation goes through an unusual outer-sphere nucleophilic substitution of the alkenes to the allyl-Rh species instead of migration insertion, and the generated carbon cation then forms the C − F bond with tetrafluoroborate as a fluoride shuttle.

Transition-metal (TM) catalyzed cross-coupling reactions are universally acknowledged as a powerful toolbox for creating carbon–carbon bond and carbon–heteroatom bond[1–4]. Among them, the Tsuji-Trost allylic substitution reactions (ASR) constitute one of the most intensively studied approaches in a controlled chemo-, regio-, and stereoselective fashion, thereby playing a crucial role in modern synthetic organic chemistry[5–7]. Consequently, they have been widely utilized in assembling structurally complicated molecules, such as natural products and bioactive molecules[8,9]. In the context of TM-catalyzed ASR, the conventional paradigm for categorizing the nucleophilic attack mode on allyl-metal intermediates is grounded in the p$K$a of the pronucleophile[10]. Mechanistic divergences are clearly observed between soft nucleophiles and hard nucleophiles (Fig. 1A). Generally, soft nucleophiles (p$K$a < 25, stabilized nucleophiles) externally attack the carbon atom of the π-allyl-metal intermediate outside the coordination sphere of the metal to directly produce the products (Fig. 1A, right)[11]. Conversely, hard nucleophiles (p$K$a > 25, unstabilized nucleophiles) initiate their attack at the metal centre leading to the

formation of an intermediate with the nucleophile bound to the metal centre, subsequently undergoing reductive elimination to yield the products (Fig. 1A, left)[12].

Alkenes are highly versatile and readily available feedstocks, representing the most attractive starting materials for the enhancement of molecular complexity and diversity[13–17]. Of particular importance is the ASR of alkenes between allylic electrophiles and alkenes, as it allows for the production of synthetically valuable diene compounds[18,19]. In this context, a few carefully tailored catalytic systems based on Rh[20,21], Pd[22–24], Ir[25,26], and Ni[27] have been used for this transformation, with alkenes generally attacking the allyl-metal species through a sequence of migratory insertion and β-H elimination within the inner-sphere mechanism (Fig. 1B, left). Despite these advancements in this area, the development of ASR of alkenes has lagged behind, attributable in part to the intrinsic predilection for an inner-sphere mechanism, which presents obstacles in tolerating multi-substituted alkenes due to the high demand on the steric hindrance in the migratory insertion process. Additionally, the formation of an

[1]West China School of Public Health and West China Fourth Hospital, West China-PUMC C.C. Chen Institute of Health, and State Key Laboratory of Biotherapy, Sichuan University, Chengdu 610041, China. [2]School of Chemistry and Chemical Engineering, Key Laboratory of Colloid and Interface Chemistry, Ministry of Education, Shandong University, Jinan 250100, China. ✉e-mail: ganglu@sdu.edu.cn; xiayingscu@scu.edu.cn

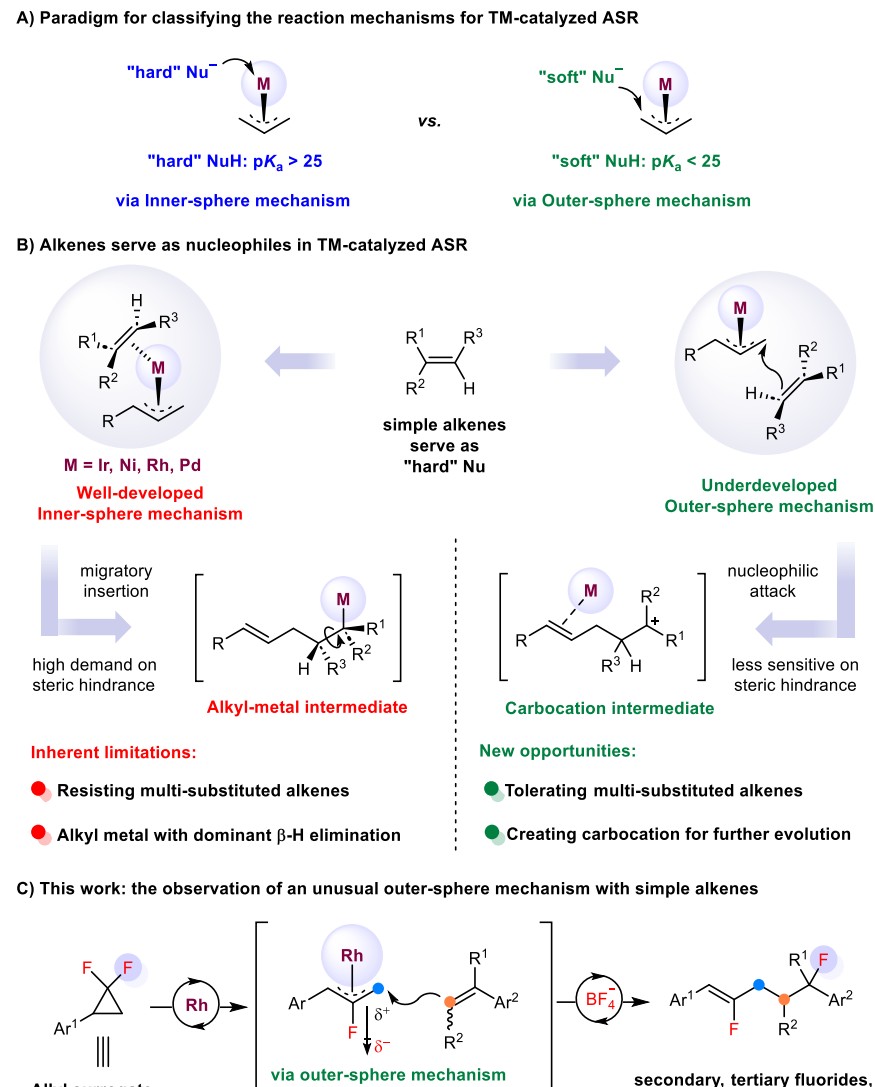

**Fig. 1 | The state-of-the-art of allylic substitution of alkenes. A** Paradigm for classifying the reaction mechanism for TM-catalyzed ASR. **B** Reaction mechanisms using alkenes as nucleophiles in TM-catalyzed ASR. **C** The observation of an unusual outer-sphere mechanism with simple alkenes. TM transition-metal, ASR allylic substitution reaction.

alkyl-metal complex through migratory insertion of the alkene is prone to terminate with a β-H elimination process, hampering further evolution of these reactions. Comparatively, the outer-sphere pathway offers greater flexibility for the alkenes attack than the inner-sphere pathway, and the resulting carbocation complexes can be further functionalized, ultimately leading to the allylic bifunctionalization of alkenes (Fig. 1B, right). To our best knowledge, there has been no study on the ASR of alkenes proceeding via an outer-sphere mechanism, mainly due to the prevalence of the inner-sphere mechanism for alkenes acting as hard nucleophiles. The development of such ASR of alkenes poses a challenge because alkenes, other than enolates and enamines[28,29], are difficult to soften to form stabilized nucleophiles. An alternative approach may entail increasing the electrophilicity of the allyl-metal complex, effectively lowering the lowest unoccupied molecular orbital (LUMO) energy of allyl-metal species, which could be conceived as an optimal strategy to address the aforementioned challenge.

In recent years, *gem*-difluorinated cyclopropanes (*gem*-DFCPs) have become a type of widely used substrates for the synthesis of monofluoro alkenes via ring-opening/cross-coupling reactions under transition-metal catalysis[30−36]. Our group has been devoted to the exploration of the reactivity of *gem*-DFCPs under rhodium catalysis[21,37−41]. These reactions involve a C−C bond activation and β-F elimination process, providing the key fluoroallyl-Rh intermediate marked by its heightened electrophilicity due to the electron-withdrawing effect of the fluorine and the cationic Rh. To this end, we surmised that the outer-sphere mechanism could emerge from reactions using the highly electrophilic fluoroallyl-Rh complex in the ASR of alkenes. Herein, we report a demonstration of an outer-sphere mechanism in the Rh-catalyzed ASR of simple alkenes using *gem*-DFCPs as allyl surrogates (Fig. 1C). This mechanistic mutation offers an opportunity for the fluorine recycling of *gem*-DFCPs via C−F bond cleavage/reformation[42,43], delivering allylic carbonfluorination products[44−48] beyond our previously reported Heck-type allylation[21] or cycloaddition[39]. Just as the advantages of the outer-sphere pathway, this ASR protocol can indeed tolerate a wide range of simple alkenes, even trisubstituted ones, enabling the synthesis of secondary, tertiary fluorides, and *gem*-difluorides with 100% atom-economy under mild

**Table 1 | Optimization of reaction conditions[a]**

X-ray of Rh(CO)$_2$(BINAP)BF$_4$
CCDC: 2307402

| Entry | Variations | Yield[b] |
|---|---|---|
| 1 | None | 92% (88%)[c] |
| 2 | w/o Rh(CO)$_2$(BINAP)BF$_4$ | 0 |
| 3 | [Rh(CO)$_2$Cl]$_2$, BINAP, and AgBF$_4$ instead of Rh(CO)$_2$(BINAP)BF$_4$ | 0–80% |
| 4 | Rh(CO)$_2$(dppf)BF$_4$ instead of Rh(CO)$_2$(BINAP)BF$_4$ | 0 |
| 5 | Rh(CO)$_2$(dppbz)BF$_4$ instead of Rh(CO)$_2$(BINAP)BF$_4$ | trace |
| 6 | Rh(CO)$_2$(BIPHEP)BF$_4$ instead of Rh(CO)$_2$(BINAP)BF$_4$ | 18% |
| 7 | Rh(COD)(BINAP)BF$_4$ instead of Rh(CO)$_2$(BINAP)BF$_4$ | trace |
| 8 | 1,4-Dioxane instead of PhCl | 36% |
| 9 | PhCF$_3$ instead of PhCl | 58% |
| 10 | PhF instead of PhCl | 64% |
| 11 | 40 °C instead of 50 °C | 42% |
| 12 | 16 h | 70% |

[a]Reaction was performed on 0.1 mmol scale. [b]Yield was determined by [1]H NMR using 1,1,2,2-tetrachloroethane as the internal standard. [c]Isolated yield. BINAP, 1,1'-binaphthyl-2,2'-diphenyl phosphine; dppf, 1,1'-bis(diphenylphosphino)ferrocene; dppbz, 1,2-bis(diphenylphosphino)benzene; BIPHEP, 2,2'-bis(diphenylphosphino)-1,1'-biphenyl; COD, 1,5-cyclooctadiene.

conditions. Mechanistically, we conducted experimental and theoretical investigations of the mechanism of this irregular outer-sphere mechanism in allylation chemistry.

## Results

### Reaction optimization
Our initial investigation used the readily available (2,2-difluorocyclopropyl)benzene **1a** and styrene **2a** as the substrates (Table 1). In our prior work on site-divergent fluoroallylation of alkenes[21], the use of monodentate phosphine ligand favored an inner-sphere mechanism, resulting in two regioisomeric 1,4-dienes. Building upon this, employing the bidentate phosphine ligands led to the coordination sphere of rhodium being more stringent, which somewhat helps the ASR of alkenes to resist the inner sphere pathway. Optimization revealed that this reaction is conducted best with 4 mol% [Rh(CO)$_2$(BINAP)BF$_4$] as the pre-catalyst in PhCl at 50 °C for 24 h, which produces the allylic carbofluorination product **3a** in 88% isolated yield (Entry 1). In the absence of the rhodium catalyst, this reaction didn't proceed (Entry 2). We also investigated the use of a cationic Rh catalyst generated in situ from [Rh(CO)$_2$Cl]$_2$, BINAP, and AgBF$_4$ in PhCl (Entry 3). Intriguingly, a dramatic but repeatable difference in the catalytic efficiency of rhodium catalysts pre-mixed by different feeding sequences was observed (for more details, please refer to Supplementary Table 2). The observation of the results in entry 3 prompted us to use [Rh(CO)$_2$(BINAP)BF$_4$] as the pre-catalyst to achieve steady experimental results and improve the operability. The structure of [Rh(CO)$_2$(BINAP)BF$_4$] was unambiguously verified by X-ray analysis. Replacing BINAP with other types of bidentate phosphine ligands, such as dppf, dppbz, and BIPHEP, led to low yields of **3a** (Entries 4-6). Additionally, [Rh(COD)(BINAP)BF$_4$] was found to be much less effective with low conversion (Entry 7), indicating that the presence of carbon monoxide ligands is crucial for the success of the reaction. Furthermore, we also investigated the solvent effect and found that PhCl was superior to other similar solvents such as PhCF$_3$ and PhF, while more polar solvents like 1,4-dioxane resulted in significantly lower yield (Entries 8–10). Decreasing the reaction temperature and

shortening the reaction time led to worse results with incomplete conversion of the substrates (Entries 11-12).

### Substrate scope
Having identified the optimized reaction conditions, we proceeded to investigate the scope of this allylic carbofluorination reaction (Fig. 2). Firstly, we examined a range of alkenes with *gem*-DFCP **1a** as the coupling partner. As shown in Scheme 2, the styrene derivatives bearing electron-withdrawing groups such as halogen (**3b, 3c**) and ester (**3d**) afforded the desired products in moderate yields. 1,1-Disubstituted alkenes containing various substitutions were compatible with this transformation, providing the corresponding tertiary fluorides in good yields (**3e–3l**), in which the structure was confirmed by X-ray analysis in the case of **3g**. In some substrates (**3e, 3i**, and **3l**), the Heck-type allylation becomes the major pathway under the standard reaction conditions, and it was found that lowering the reaction temperature to 40 °C can make the carbofluorination return predominate.

Furthermore, 1,2-disubstituted alkenes also proceed smoothly under this catalytic system. Substrates decorated with methyl, propyl, benzyl, and functionalized alkyl substituents containing terminal halogen, ester, and ether group were tolerated and afforded **3m–3t** in good yields but with low diastereocontrol. The pure *Z*-alkene **2m** also underwent this transformation to yield the product **3m** in a decreased yield, and the same dr. However, achieving high conversion of the *Z*-alkene necessitated a longer reaction time. Note that these 1,2-disubstituted alkenes exhibited high reactivity in our reported [3 + 2] cycloaddition reaction[39], reflecting that the current catalytic system displayed good chemoselectivity despite the observation of a small number of cycloaddition products (<5%) in most case of 1,2-disubstituted alkenes. The challenging trisubstituted alkenes can undergo this transformation, yielding a diaryl tertiary fluoride in an acceptable yield (**3v**), which was difficult to occur through the inner-sphere pathway. Remarkably, phenylsubstituted cyclohexene also functioned as a suitable reaction substrate in this Rh-catalyzed carbofluorination, and the target product (**3w**) was accessible in 51% yield as a single diastereoisomer, in which

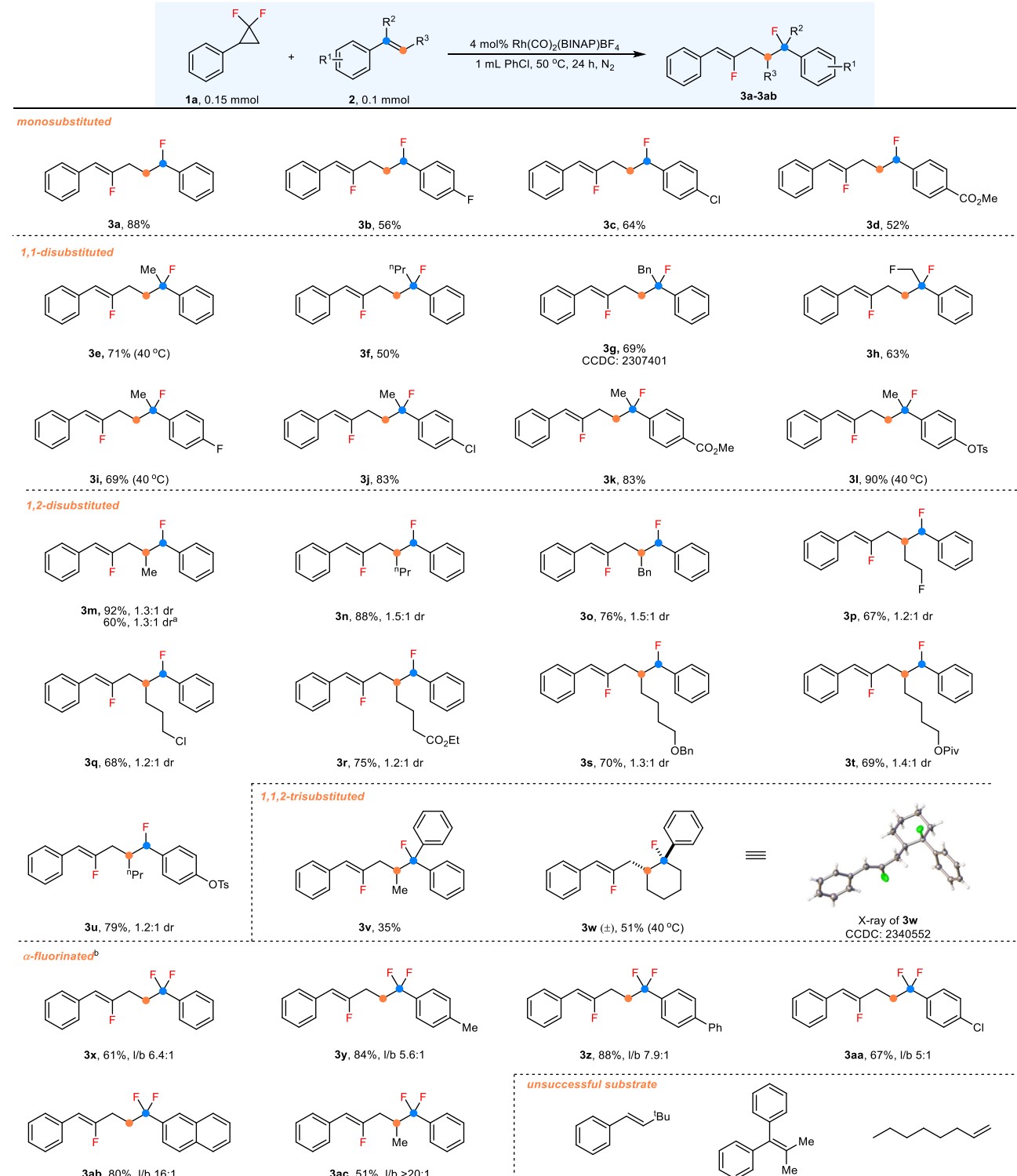

**Fig. 2 | Substrate scope of alkenes.** Reaction conditions: **1a** (0.15 mmol), **2a** (0.1 mmol), [Rh(CO)₂(BINAP)BF₄]₂ (4 mol%) in PhCl (1 mL) at 50 °C for 24 h. All yields are the average of three runs. ᵃThe reaction was conducted at 50 °C for 48 h using pure *Z*-alkene. ᵇThe ratio of l/b refers to the allyl regioselectivity (linear/branched).

the relative configuration was unambiguously confirmed as *cis*-configuration by X-ray crystallography.

To further showcase the diversity of the product synthesized by this protocol, α-fluorostyrenes were employed in this reaction, which provide facile access to *gem*-difluorinated compounds. α-Fluorostyrene itself (**3x**) or substrates with functional groups encompassing methyl (**3y**), phenyl (**3z**), and chlorine (**3aa**) at the *para*-position led to the formation of *gem*-difluorinated products

efficiently with moderate regioselectivity. It is worth noting that 2-naphthyl and 1,2-disubstituted α-fluoroalkene were tolerated as well, delivering products **3ab** and **3ac** in good yields with good regioselectivity. Very low conversions were observed in the reactions of the bulkier 2-ᵗBu-substituted styrene and tetrasubstituted olefin, likely due to the severely increasing steric hindrance. The alkyl-substituted alkene exhibited no reactivity under this current catalytic system.

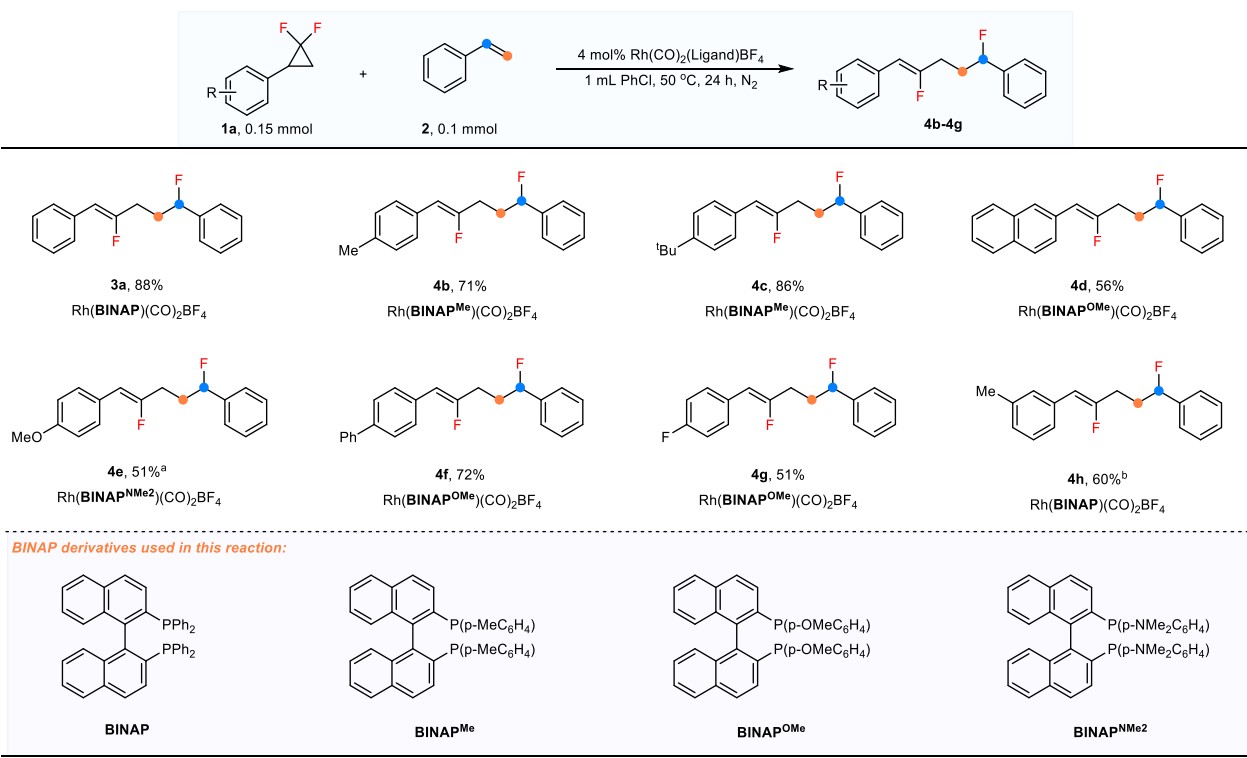

**Fig. 3 | Substrate scope of *gem*-DFCPs.** Reaction conditions: **1a** (0.15 mmol), **2a** (0.1 mmol), [Rh(CO)₂(BINAP)BF₄]₂ (4 mol%) in PhCl (1 mL) at 50 °C for 24 h. All yields are the average of three runs. [a]48 h. [b]PhCF₃ as the reaction solvent.

After that, the reactivity of *gem*-DFCPs was evaluated with styrene as the partner (Fig. 3). Initially, we performed experiments using a series of *gem*-DFCPs under standard conditions, but the results were unsatisfactory. The substituted *gem*-DFCPs lead to two outcomes: either a high conversion without the formation of the desired products or a low conversion resulting in the recovery of the starting substrates. These results indicate that this reaction is sensitive to the electronic and steric hindrance effects of the substituents on *gem*-DFCPs. While substituted *gem*-DFCPs did not participate in the allylic carbofluorination reaction using Rh(CO)₂(BINAP)BF₄ as the pre-catalyst in our preliminary attempts, it was found that the rhodium pre-catalyst synthesized from BINAP derivatives could facilitate the formation of product in certain substrate cases. When Rh(CO)₂(BINAP^Me)BF₄ was employed as the pre-catalyst, substrates bearing methyl (**4b**) and tert-butyl substituents (**4c**) provided the target products in good yields. Additionally, 2-naphthyl *gem*-DFCP works well using Rh(CO)₂(BINAP^OMe)BF₄ as the catalyst (**4d**). *gem*-DFCP with a strong electron-donating group (OMe) required a more electron-rich ligand and longer reaction time (**4e**). Rh(CO)₂(BINAP^OMe)BF₄ is also an efficient pre-catalyst for fluoro- and phenyl-substituted substrates (**4f**, **4g**). Furthermore, the choice of reaction solvent is sometimes important for the transformation, with *meta*-methyl substituted substrate exhibiting good reactivity and yielding the product in good yield when PhCF₃ was used as the solvent (**4h**). In general, the reaction efficiency is very sensitive to the variation of *gem*-DFCPs, which is uncommon in transition-metal catalysis.

## Synthetic application
To explore the synthetic application of this protocol, we conducted the model reaction on a gram-scale, resulting in **3a** in 81% isolated yield (Fig. 4A). In addition, the benzyl C–F bond in the product can undergo defluorinative coupling through hydrogen bonding with the use of HFIP as a hydrogen bond donor[49]. Friedel-Crafts reaction of **3a** with *p*-xylene generated a C–C bond coupling product **5a** in 56% yield.

Treatment of **3a** with other types of nucleophiles forms C–C, C–N, and C–O bonds efficiently (Fig. 4B). Furthermore, the Kumada-coupling between **5c** with Grignard reagents as the nucleophiles gave the diaryl olefin **6a** in 76% yield (Fig. 4C). When methyl 4-phenylpent-4-enoate (**2ac**) was subjected to the standard reaction conditions, it successfully yielded compound **7a**, a lactone, in 56% yield (Fig. 4D). This represents another example of evolution facilitated by an outer-sphere mechanism in TM-catalyzed ASR with alkenes. Taking together, our developed method not only offers general access to realize C/F olefin bifunctionalization, but also is able to achieve formal C/C, C/N, and C/O bifunctionalization of alkenes through post-transformation of the fluoride products.

## Mechanistic studies
Initially, we indeed considered the possibility of inner-sphere mechanism in this transformation, and a sequence of migratory insertion and reductive elimination of C–F bond finishes the product (Fig. 5A). However, the poor diastereoselectivity (around 1:1, see Fig. 2) in the allylic carbofluorination of 1,2-disubstituted alkenes excludes the possibility of olefin migratory insertion/C–F reductive elimination sequence. When subjecting (3-methylbut-1-en-2-yl)benzene **2aj** as the substrate to the standard reaction conditions, we observed the formation of benzyl fluoride and tertiary alkyl fluoride in a ratio of 1:1 (Fig. 5B). This result indicates that a carbon cation intermediate is involved in the allylic carbofluorination reaction process. We also investigated the influence of the counter-anion in the rhodium pre-catalysts and found that only tetrafluoroborate is able to ensure the progress of the reaction (Fig. 5C).

## DFT calculations
To better understand the reaction mechanism, especially regarding how the C–C and C–F bonds are formed, DFT calculations are then conducted. The computed energy profile is shown in Fig. 6. Among a series of possible active cationic Rh(I) catalyst species under the

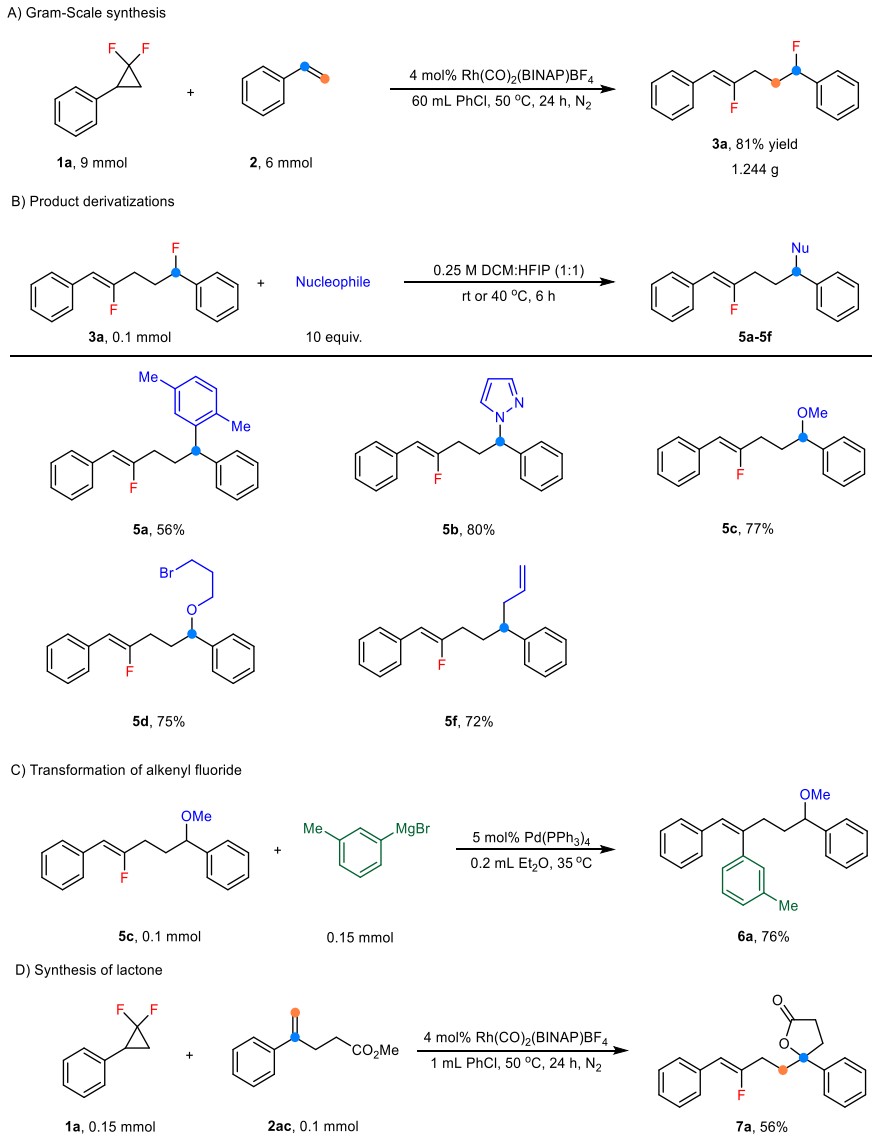

**Fig. 4 | Synthetic applications. A** Gram-Scale synthesis. **B** Product derivatization. **C** Transformation of alkenyl fluoride. **D** Synthesis of lactone.

experimental condition, **Int1** is the most stable structure and was chosen as the zero-energy reference (for more details, please refer to page 45 of the Supplementary information file). The reaction begins with the oxidative addition of *gem*-DFCP **1a** via **TS1** with a barrier of 24.4 kcal/mol. The formed four-membered rhodacycle **Int2** undergoes β-F elimination (**TS2**), which gives the fluoroallyl-Rh(III) intermediate **Int3**. Although it is straightforward for styrene to be inserted into **Int3** via the inner-sphere mechanism, this process is less possible due to the high barrier (**TS3**, $\Delta G^{\ddagger} = 40.8$ kcal/mol). Alternatively, we considered the possibility of outer-sphere attack of a nucleophile with allyl-Rh(III) species[50–53]. The computed transition state (**TS4**) requires a barrier of 24.1 kcal/mol, which is much lower than that of styrene migratory insertion (**TS3**). This outer-sphere nucleophilic attack generates the Rh(I) intermediate **Int5**, in which the rhodium center is coordinated by the double bond and a formal benzylic carbocation can be observed. The formation of a carbon cation is well consistent with the experimentally observed fluoride rearrangement (Fig. 5B) and poor diastereoselectivity in the reaction of 1,2-disubstituted alkenes (Fig. 2). Subsequently, the benzylic carbocation can be fluorinated by $BF_4$ anion[54–56], which is a barrierless process (for more details, please refer to Supplementary Fig. 9), to afford the allylic carbofluorination product of styrene (**3a**). The fluoride abstraction of Rh(I)–F (**Int7**) by in situ

formed $BF_3$ regenerates the active catalyst species. This $BF_4^-/BF_3$ cycle emphasizes that, compared with other counter anions (Fig. 5C), the $BF_4$ anion plays a crucial role in enabling the reaction rather than serving as a spectator counter anion[47,57].

## Proposed mechanism
Based on both experimental and computational results, an overall catalytic reaction mechanism is given in Fig. 7. Firstly, the oxidative addition of *gem*-DFCP **1a** with the active Rh(I) catalyst gives the intermediate **A**, which then undergoes β-F elimination to deliver the fluoroallyl Rh species **B**. Subsequently, the outer-sphere nucleophilic attack of styrene **2a** on intermediate **B** can generate a benzylic cation **C**. This carbocation **C** finally leads to the production of fluoride **3a** by abstracting a fluoride from $BF_4^-$, along with the formation of one molecular $BF_3$. The resulting $BF_3$ can react with the Rh(I)–F species to form $BF_4^-$ and regenerate the rhodium catalyst.

## Discussion
In conclusion, we have introduced an initial application of an outer-sphere mechanism in the Rh-catalyzed ASR of simple alkenes using *gem*-DFCPs as allyl surrogates. In comparison to the conventional inner-sphere pathway, this approach shows greater tolerance of

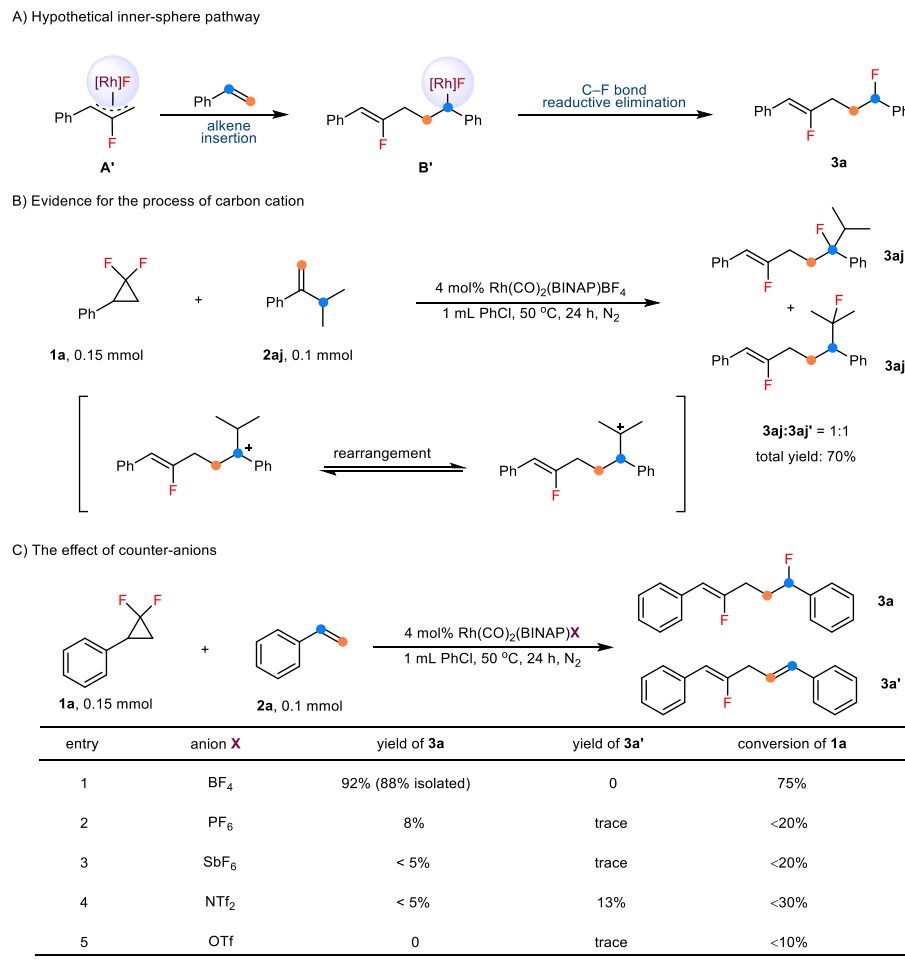

**Fig. 5 | Preliminary control experiments. A** Hypothetical inner-sphere pathway. **B** Evidence for the process of carbon cation. **C** The effect of counter-anions.

| entry | anion **X** | yield of **3a** | yield of **3a'** | conversion of **1a** |
|-------|-------------|-----------------|------------------|----------------------|
| 1 | BF₄ | 92% (88% isolated) | 0 | 75% |
| 2 | PF₆ | 8% | trace | <20% |
| 3 | SbF₆ | < 5% | trace | <20% |
| 4 | NTf₂ | < 5% | 13% | <30% |
| 5 | OTf | 0 | trace | <10% |

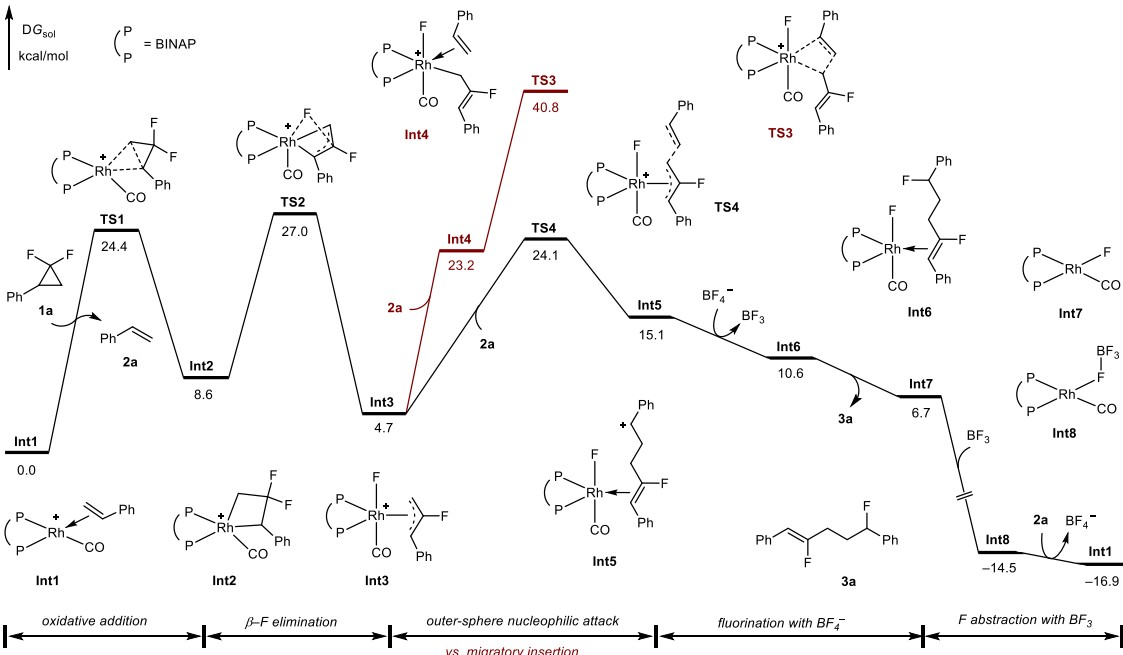

**Fig. 6 | Computational studies.** DFT-computed pathways for alkenes carbofluorination with *gem*-DFCPs.

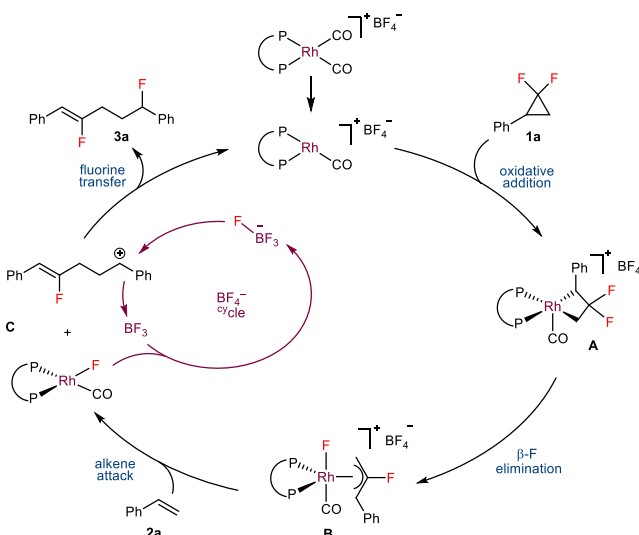

**Fig. 7 | Proposed overall catalytic cycle.** Reaction mechanism for rhodium-catalyzed allylic fluorination of simple alkenes.

alkene substrates, providing access to various benzyl fluorides and *gem*-difluorides with 100% atom economy. The bifunctional reactivity of *gem*-DFCPs, enabled by the outer-sphere mechanism, sets it apart from the prevalent defluorination coupling reaction. Furthermore, the practicality of this transformation was demonstrated by convenient C–F coupling to form new C–C, C–N, and C–O bonds. Mechanistic studies suggest that the C–C bond forms via an unusual outer-sphere allylic nucleophilic substitution with simple olefins as nucleophiles, while the C–F recombination goes through the trap of the generated carbon cation with tetrafluoroborate as a fluoride shuttle. Finally, the distinctive reaction mode catalyzed by the well-defined cationic dicarbonyl rhodium complex towards allylic carbofluorination process via outer-sphere mechanism, distinguishing defluorinative Heck-type allylation and cycloaddition in the reaction of *gem*-DFCPs with alkenes, will provide more inspirations on allylation chemistry and also transition-metal catalysis.

## Methods

### General procedure for carbofluorination of alkenes

In a nitrogen-filled glove box, a 4 mL vial equipped with a stir bar was charged with [Rh(CO)$_2$(BINAP)BF$_4$] (3.5 mg, 0.004 mmol, 4 mol%) and PhCl (1 mL). After stirring at room temperature for about 5 min, *gem*-DFCPs **1** (0.15 mmol) and the corresponding alkene **2** (0.1 mmol) were added to the resulting yellow catalyst solution. The 4 mL vial was sealed and removed from the glove box and stirred at 50 °C for 24 h. The reaction mixture was cooled to room temperature and purified by chromatography on a silica gel column to give the fluorides **3** or **4**.

## Data availability

The NMR, HRMS, and DFT computation data generated in this study are provided in the Supplementary Information. Cartesian coordinates of the calculated structures are available from Supplementary Data 1. The X-ray crystallographic coordinates for structures reported in this study have been deposited at the Cambridge Crystallographic Data Centre (CCDC), under deposition numbers **3g** (CCDC 2307401), **3w** (CCDC 2340552) and Rh(CO)$_2$(BINAP)BF$_4$ (CCDC 2307402). These data can be obtained free of charge from The Cambridge Crystallographic Data Centre via www.ccdc.cam.ac.uk/data_request/cif. All data are available from the corresponding author upon request.

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

## Acknowledgements

This work is supported by the National Natural Science Foundation (Grants 22371189, 22371171) and the "Thousand Young Talents Program of China" (Grant 15-YINGXIA). Dr. Xuejiao Song, Dr. Xunxun Deng, and Dr. Tianli Zheng in Public Health and Preventive Medicine Provincial Experiment Teaching Center at Sichuan University are acknowledged for their support for instrument management.

## Author contributions

Y.Z.(Yaxin) and Y.X. conceived and designed the experiments. Y.Z.(Yaxin), Z.-T.J., Y.Z.(Yulei), J.C. and H.Z. performed the experiments,

compound characterization, and data analysis. H.G. and G.L. performed the DFT calculations. Y.Z.(Yaxin), H.G., G.L. and Y.X. co-wrote the paper.

## Competing interests

The authors declare no competing interests.
