## [Peer Review File · Nature Communications]

Observation of unusual outer-sphere mechanism using simple alkenes as nucleophiles in allylation chemistryREVIEWER COMMENTS

Reviewer #1 (Remarks to the Author):

In this paper, Xia and coworkers introduced a novel allyl coupling method combining alkenes with gem-difluorinated cyclopropanes. This reaction showcases a wide range of styrene substrates, achieves 100% atom economy, and operates under mild conditions. The resulting products including secondary, tertiary fluorides, and gem-difluorides, which are intriguing and typically challenging to synthesize in a single step using traditional methods. Notably, this Rh-catalyzed study reveals bifunctional reactivity with the same alkene substrates, diverging from the team's prior research efforts (see ref. 21 and 37). The successful scale-up synthesis and product derivatization underscore the potential practical applications of this protocol.

Combining mechanistic experiments with DFT studies, the authors clarified that the initial allylic substitution of alkenes proceeds through an outer-sphere mechanism. Subsequent C-F bond formation involves fluorine rebounding with BF₄⁻ acting as a fluoride shuttle. This study not only unveils a distinctive reaction mechanism in alkene allylation chemistry but also paves the way for the allylic bifunctionalization of alkenes. In summary, I found this study to be both intriguing and well-structured. I recommend its publication in Nature Communications after minor revisions. Below are some issues that should be addressed before publication.

- 1) In Figure 1c, the wavy bond in the alkenes indicates that both cis- and trans-configurations of olefins are viable in this reaction. However, I could not find any information regarding the impact of the configuration of 1,2-disubstituted olefins on this reaction. Can the authors give more explanation on this?
- 2) Regarding the scope, while this approach demonstrates compatibility with various simple alkenes, it primarily emphasizes styrene derivatives. Have the researchers explored alkyl, cyclic alkenes, or dienes, and if so, what were the outcomes? Additionally, for 1,2-disubstituted alkenes, is it possible for alkenes with bulkier substitutions like *i*Pr or *t*Bu to yield the desired product?
- 3) As illustrated in Figure 3, the range of gem-DFCPs is somewhat restricted. Presumably, the authors have thoroughly explored this limitation. According to the authors, gem-DFCPs with varying substitutions often necessitate the use of specific ligands to achieve the desired outcome; otherwise, they may result in either low conversion or high conversion without the intended product, an atypical occurrence. It appears that more electron-rich substrates may demand equally electron-rich ligands. Could electron-deficient gem-DFCPs potentially be viable with electron-deficient BINAP derivatives? Furthermore, I recommend that the outcomes of unsuccessful substrates be compiled in the Supplementary Information to provide readers with a deeper understanding and insights.
- 4) In Figure 1a, I suggest that the authors modify the notation for "hard" Nu: pK_a > 25 as "hard" NuH: pK_a > 25; and "soft" Nu: pK_a < 25 as "soft" NuH: pK_a < 25.
- 5) I also recommended that the authors contemplate using different colors to delineate the structural motifs and highlight the bond formations, especially in figure 2 and 3.

Reviewer #2 (Remarks to the Author):

This manuscript by Lu and Xia et al. describes the development and mechanistic investigation of a Rh-catalyzed allylic substitution reaction of alkenes using gem-difluorocyclopropanes as allyl surrogates. The authors demonstrate that alkene products with fluorination at both sp² and sp³ centers can be obtained through this protocol, which

provide synthetic handles that can be further elaborated in a variety of ways. Mechanistic experiments and DFT calculations lent support to an outer-sphere C–C formation mechanism producing a carbocation, which is captured by fluoride to achieve high atom economy for the overall transformation. The reaction is both interesting from a mechanistic standpoint and synthetically useful, and should be considered for publication in Nature Communications provided that the following issues are addressed in a revision:

1. Though a trisubstituted alkene (3v) was included in the scope, the yield is much lower. Can the authors comment on why that is? Is it due to side reactions or lack of reactivity? Also, are tetra-substituted alkenes not reactive at all? These are important limitations that the audience would want to know about before they apply the chemistry.
2. The DFT optimizations were performed with B3LYP, which does not account for dispersive effects. No empirical dispersion was included, either. Given that both coupling partners have aryl rings on them and the BINAP ligand has many aryl rings as well, there is a very good chance for pi-pi stacking between the ligand and substrates, so the lack of dispersion can become a serious issue that affects geometry optimization since pi-pi stacking is not sufficiently accounted for. I think the authors should at least re-optimize structures along the Int3→TS4→Int5 and Int4→TS3 pathways using B3LYP-D3(BJ) and re-evaluate the free energies to ensure that the inclusion of dispersive effects does not change the overall conclusion.
3. The sensitivity of the reaction to substitution on the gem-difluorocyclopropane is a bit puzzling. Kudos to the authors on finding the appropriate BINAP derivatives to make them work, but it would also be nice to see some explanation as to why. It appears the BINAP derivatives with more electron-donating character was necessary to promote the reaction. I imagine this may have to do with the kinetics of the oxidative addition and beta-fluoride elimination steps given their higher barriers. Can the authors provide a rationalization from a computational standpoint?
4. In the calculated energy diagram, the fluoride stays on the Rh center during the C–C formation process. Have the authors calculated the alternative pathway where the fluoride falls off of the Rh center (e.g. through reacting with BF₃) after beta-fluoride elimination? I understand that this would create a 2+ Rh complex and may be energetically unfavorable, but if the fluoride can dissociate under the reaction conditions it would significantly impact the electronic and steric character of the Rh center, and possibly change the favorability of the inner- versus outer-sphere pathways as well.

Reviewer #3 (Remarks to the Author):

This manuscript by Xia and co-workers reports a unique outer-sphere mechanism for simple alkenes in allylation chemistry, realizing a catalytic carbofluorination of alkenes with gem-difluorinated cyclopropanes. It appears that the key to success of this transformation is the use of an unconventional pre-catalyst, Rh(CO)₂(BINAP)BF₄, which steers the carbofluorination transformation via allylic substitution/fluorination sequence, sidestepping the Heck-type allylation and cycloaddition reported by the authors' group. This protocol accommodates a wide range of alkenes, even tri-substituted ones, yielding mono-fluorides and gem-difluorides with 100% atom-economy under mild and simple conditions. The synthetic practicality was also demonstrated by gram-scale experiments and product derivatizations. Mechanistic experiments and DFT calculation rationalized the intriguing out-

sphere mechanism of alkenes in allylic substitution and also how the C-F bond is formed. Overall, I fully recommend this work publication on Nature Communications, considering the following a few minor remarks addressed by the authors.

(1) It is reasonable that the stereocenter on the α -position (the position is regarding to the substrate aryl olefin) cannot be controlled as the C-F bond forms via a fluorine transfer to a carbon cation intermediate. However, as the C-C bond forms via outer-sphere allylic attack in which the Rh/Ligand is involved, readers would be interested in whether the stereocenter on the β -position can be controlled, in other words, how about the enantioselectivity of 1,2-disubstituted olefins by utilizing chiral ligands despite the low diastereoselectivity.

(2) Apart from the 1,2-disubstituted E-alkenes, how about the reactivity of Z-alkenes? The impact of the configuration of 1,2-disubstituted alkenes, including the reactivity and selectivity, warrants further investigation, which may also help understand the mechanism.

(3) In comparison to the pre-mixed catalyst solution, it is interesting that the reactivity of this well-defined catalyst, $\text{Rh}(\text{CO})_2(\text{BINAP})\text{BF}_4$, is dramatically different in this reaction. When synthesizing this type of complex for research purposes, authors should take into account factors such as stability, storage conditions, and operational environment. This essential information is suggested to be provided to facilitate prompt utilization by readers who are interested in.

(4) There is a recent transformation of gem-difluorinated cyclopropanes suggested to be cited, S. Qi, Y. Hua, L. Pan, J. Yang, J. Zhang, *Chin. J. Chem.* 2024, 42, 823-828.

Re: referee 1

(a) Original comments: *“In this paper, Xia and coworkers introduced a novel allyl coupling method combining alkenes with gem-difluorinated cyclopropanes. This reaction showcases a wide range of styrene substrates, achieves 100% atom economy, and operates under mild conditions. The resulting products including secondary, tertiary fluorides, and gem-difluorides, which are intriguing and typically challenging to synthesize in a single step using traditional methods. Notably, this Rh-catalyzed study reveals bifunctional reactivity with the same alkene substrates, diverging from the team’s prior research efforts (see ref. 21 and 37). The successful scale-up synthesis and product derivatization underscore the potential practical applications of this protocol. Combining mechanistic experiments with DFT studies, the authors clarified that the initial allylic substitution of alkenes proceeds through an outer-sphere mechanism. Subsequent C-F bond formation involves fluorine rebounding with BF₄⁻ acting as a fluoride shuttle. This study not only unveils a distinctive reaction mechanism in alkene allylation chemistry but also paves the way for the allylic bifunctionalization of alkenes. In summary, I found this study to be both intriguing and well-structured. I recommend its publication in Nature Communications after minor revisions. Below are some issues that should be addressed before publication.”*

Thanks for the reviewer’s positive comments and constructive suggestions, and we have addressed the following questions to improve our manuscript.

(b) Original comments: *“In Figure 1c, the wavy bond in the alkenes indicates that both cis- and trans-configurations of olefins are viable in this reaction. However, I could not find any information regarding the impact of the configuration of 1,2-disubstituted olefins on this reaction. Can the authors give more explanation on this?”*

Thank the reviewer to point out it. We have supplemented additional experimental results regarding the effect of the configuration of 1,2-disubstituted alkenes at different ratio of *E/Z* under otherwise the same conditions. (see below). When using 1:1 *E/Z* mixture of alkene **2m**, the yield of **3m** was decreased to 75% with 1.5:1 dr. Treating the pure *Z*-**2m** under standard conditions resulted in a 50% yield of product **3m** with a 2.0:1 dr and incomplete conversion. Prolonging the reaction time to 48 hours increased the yield of **3m** to 66% with a comparable dr value. These results have been included in the revised supplementary information, which was briefly mentioned in the revised manuscript.

entry	alkene	time	yield of 3a ^a	dr
1	E - 2m	24 h	92%(88% ^b)	1.3:1
2	2m (E : Z 1:1)	24 h	75%	1.5:1
3	Z - 2m	24 h	50%	2.0:1
4	Z - 2m	48 h	66%(60% ^b)	1.3:1

^aYield was determined by ¹H NMR using the 1,1,2,2-tetrachloroethane as the internal standard.

^bIsolated yield

(c) **Original comments:** “Regarding the scope, while this approach demonstrates compatibility with various simple alkenes, it primarily emphasizes styrene derivatives. Have the researchers explored alkyl, cyclic alkenes, or dienes, and if so, what were the outcomes? Additionally, for 1,2-disubstituted alkenes, is it possible for alkenes with bulkier substitutions like *i*Pr or *t*Bu to yield the desired product?”

Thank the reviewer’s question! We have conducted tests on the reactivity of the alkenes mentioned above. Alkyl alkenes do not exhibit reactivity, and dienes like 1,3-dienes did not provide the expected carbonfluorination products. Furthermore, we have also investigated the effect of steric hindrance by examining *i*Pr or *t*Bu substituted 1,2-alkenes. The *i*Pr-substituted one gave the carbonfluorination products in only 8% yield, while the *t*Bu-substituted one did not show reactivity. These results suggested that the nucleophilic attack of alkenes is significantly impeded by the bulky substitutions in this outer-sphere mechanism. Additionally, the tetra-substituted alkene is not reactive under this catalytic system, similarly due to the increasing steric hindrance. In the case of cyclic alkenes, 2,3,4,5-tetrahydro-1,1'-biphenyl underwent this carbonylation to give tertiary fluoride **3w** in 51% yield with a *cis*-configuration, albeit with the formation of Heck-type products with 25% yield. These experimental results have been included in the revised manuscript.

(d) Original comments: “As illustrated in Figure 3, the range of gem-DFCPs is somewhat restricted. Presumably, the authors have thoroughly explored this limitation. According to the authors, gem-DFCPs with varying substitutions often necessitate the use of specific ligands to achieve the desired outcome; otherwise, they may result in either low conversion or high conversion without the intended product, an atypical occurrence. It appears that more electron-rich substrates may demand equally electron-rich ligands. Could electron-deficient gem-DFCPs potentially be viable with electron-deficient BINAP derivatives? Furthermore, I recommend that the outcomes of unsuccessful substrates be compiled in the Supplementary Information to provide readers with a deeper understanding and insights.”

That’s a great question! During the exploration of *gem*-DFCPs scope, we considered synthesizing the electron-deficient BINAP derivatives. Due to the limitation in synthesis execution, only 4-fluorine substituted BINAP was successfully synthesized, leading to a comparable result to BINAP with 4-fluorobenzene substituted *gem*-DFCPs as the starting material. In addition, we have synthesized the *rac*-SegPhos derivatives containing some electron-withdrawing groups, including fluorine, 3,5-di-fluorine, chlorine, trifluoromethyl, and 3,5-di-trifluoromethyl group. Treatment of electron-deficient *gem*-DFCPs with these electron-deficient SegPhos derivatives did not significantly improve reactivity and conversion (see below). Among them, 3,5-di-fluorine substituted SegPhos derivative showed the better yield than fluorine- and chlorine-substituted ones. More electron-deficient SegPhos derivatives, such as trifluoromethyl-substituted one or 3,5-di-trifluoromethyl one did not exhibit the reactivity.

Among the failed substrates, *gem*-DFCPs containing chlorine, bromine, ester, and carbonyl group led to the yields of corresponding product below 20% with 5-20% conversions, while *gem*-DFCPs bearing strong with-drawing groups such as trifluoromethyl, cyan and nitro group, and alkyl-substituted *gem*-DFCPs have no reactivity. These failed substrates have been summarized and added in the revised supplementary information.

(e) Original comments: “In Figure 1a, I suggest that the authors modify the notation for “hard” Nu: pKa > 25 as “hard” NuH: pKa > 25; and “soft” Nu: pKa < 25 as “soft” NuH: pKa < 25.”

Thanks for the valuable comment. In accordance with the reviewer’s recommendation, we have revised it in our revised manuscript.

(f) Original comments: “I also recommended that the authors contemplate using different colors to delineate the structural motifs and highlight the bond formations, especially in figure 2 and 3.”

Thank the reviewer’s good suggestion. We have decided to use blue and orange circles to indicate the two reaction sites of alkenes in our revised manuscript, respectively, to

clearly distinguish the structure and the new bond formation on the structural motifs of all products (revised style sees below).

Re: referee 2

(a) Original comments: "This manuscript by Lu and Xia et al. describes the development and mechanistic investigation of a Rh-catalyzed allylic substitution reaction of alkenes using gem-difluorocyclopropanes as allyl surrogates. The authors demonstrate that alkene products with fluorination at both sp^2 and sp^3 centers can be obtained through this protocol, which provide synthetic handles that can be further elaborated in a variety of ways. Mechanistic experiments and DFT calculations lent support to an outer-sphere C–C formation mechanism producing a carbocation, which is captured by fluoride to achieve high atom economy for the overall transformation. The reaction is both interesting from a mechanistic standpoint and synthetically useful, and should be considered for publication in *Nature Communications* provided that the following issues are addressed in a revision:"

Thanks for the reviewer's positive comments and constructive suggestions, and we have addressed the following questions to improve our manuscript.

(b) Original comments: "Though a trisubstituted alkene (3v) was included in the scope, the yield is much lower. Can the authors comment on why that is? Is it due to side reactions or lack of reactivity? Also, are tetra-substituted alkenes not reactive at all? These are important limitations that the audience would want to know about before they apply the chemistry."

That's a great question! For trisubstituted alkene (3v), there are two by-products: Heck-type product and allylic hydroxylation product, in addition to the allylic fluorination product 3v. We guessed that the formation of the Heck-type product may be due to the advantages conferred by the thermodynamic stability of tetra-substituted alkenes. After treating Rh-catalyst under vacuum overnight, the hydroxylation products decreased somewhat to 17%, but the yield of target product 3w still remained at 35%. This outcome suggests that the hydroxylation product may have originated from H₂O, and a little of H₂O

did not seem to impact the yield of **3w**. This description has been added in the revised supplementary information.

Additionally, we introduced a new tri-substituted alkene, 2,3,4,5-tetrahydro-1,1'-biphenyl, which underwent this carbofluorination to produce the tertiary fluoride **3w** in 51% yield with a single *cis*-configuration, albeit with the formation of Heck-type products with 25% yield. For tetra-substituted alkenes, they are not reactive with low conversion under this catalytic system, probably due to the increasing steric hindrance.

(c) Original comments: "The DFT optimizations were performed with B3LYP, which does not account for dispersive effects. No empirical dispersion was included, either. Given that both coupling partners have aryl rings on them and the BINAP ligand has many aryl rings as well, there is a very good chance for pi-pi stacking between the ligand and substrates, so the lack of dispersion can become a serious issue that affects geometry optimization since pi-pi stacking is not sufficiently accounted for. I think the authors should at least re-optimize structures along the Int3—>TS4—>Int5 and Int4—>TS3 pathways using B3LYP-D3(BJ) and re-evaluate the free energies to ensure that the inclusion of dispersive effects does not change the overall conclusion."

According to the reviewer's suggestions, we reoptimized several key intermediates and transition states (**Int3**, **TS4**, **TS5**, **Int4** and **TS3**) using B3LYP-D3(BJ) functional. Although the energies of these structures differ by a few kcal/mol between B3LYP and B3LYP-D3(BJ),

the conclusion that the outer-sphere nucleophilic attack (**TS4**) is superior to migratory insertion (**TS3**) is not affected by the choice of density functionals.

Optimization method	Int3	TS4	Int5	Int4	TS3
B3LYP	4.7	24.1	15.1	23.2	40.8
B3LYP-D3(BJ)	4.9	25.1	19.2	25.6	44.8

(d) Original comments: “The sensitivity of the reaction to substitution on the gem-difluorocyclopropane is a bit puzzling. Kudos to the authors on finding the appropriate BINAP derivatives to make them work, but it would also be nice to see some explanation as to why. It appears the BINAP derivatives with more electron-donating character was necessary to promote the reaction. I imagine this may have to do with the kinetics of the oxidative addition and beta-fluoride elimination steps given their higher barriers. Can the authors provide a rationalization from a computational standpoint?”

Thank the reviewer’s good suggestion. During the reaction screening and development, we realized this reaction is sensitive to both substituents on *gem*-difluorinated cyclopropane and the electronic properties of ligands. According to the reviewer’s suggestions, we computed the barriers of oxidative addition and β -F elimination for substrate **1g** with the Rh catalysts supported by BINAP and BINAP^{OMe} ligands. As shown below, the electron-rich BINAP^{OMe} ligand can indeed lower the barrier of oxidative addition, but exert small influence on β -F elimination.

Experimentally, some diaryl bidentate phosphine ligands with electron-deficient character also were synthesized to perform the reactions of substrate **1g**. It was found that some electron-deficient ligands also enhanced the reaction efficiency to some extent. However, more electron-deficient SegPhos derivatives, such as trifluoromethyl-substituted one or 3,5-di-trifluoromethyl one did not display the reactivity. These results suggested that this sensitivity may not solely depend on the kinetic of oxidative addition

or β -F elimination process. Overall, the reaction efficiency proves highly responsive to variations in *gem*-difluorinated cyclopropanes, a rarity in transition-metal catalysis. Presently, we acknowledge that explaining this unusual phenomenon through calculations or experiments poses a significant challenge.

(e) **Original comments:** “In the calculated energy diagram, the fluoride stays on the Rh center during the C–C formation process. Have the authors calculated the alternative pathway where the fluoride falls off of the Rh center (e.g. through reacting with BF₃) after beta-fluoride elimination? I understand that this would create a 2+ Rh complex and may be energetically unfavorable, but if the fluoride can dissociate under the reaction conditions it would significantly impact the electronic and steric character of the Rh center, and possibly change the favorability of the inner- versus outer-sphere pathways as well.”

Thanks for reviewer’s great suggestion! We calculated the pathway of fluoride elimination with BF₃ and the subsequent C–C formation process. The formation of Rh²⁺ complex via fluoride abstraction by BF₃ is thermodynamically accessible, only uphill by 1.9 kcal/mol. However, even mediated by this Rh²⁺ complex (**Int9**), the outer-sphere nucleophilic attack (**TS5**) is still superior to migratory insertion (**TS6**). Although the Rh center in **Int9** has an empty site for styrene insertion, the allyl moieties in **TS6** still adopts η^1 geometry to release steric repulsions with the bidentate phosphine ligand, thus leading to a higher barrier. Compared with the outer-sphere nucleophilic attack (**TS4**, Fig. 6) and migratory insertion (**TS3**, Fig. 6) mediated by the Rh⁺ complex, these transition states with the Rh²⁺ complex (**TS5** and **TS6** shown below) require lower barriers. However, the feasibility of this alternative Rh²⁺-mediated pathway depends on the concentration of free BF₃ in the experimental condition. Because BF₃ is *in situ* formed in the Rh⁺ pathway and

consumed to complete the catalytic cycle, the Rh^{2+} species generated from fluoride abstraction by BF_3 is less possible.

Re: referee 3

(a) Original comments: "This manuscript by Xia and co-workers reports a unique outer-sphere mechanism for simple alkenes in allylation chemistry, realizing a catalytic carbofluorination of alkenes with gem-difluorinated cyclopropanes. It appears that the key to success of this transformation is the use of an unconventional pre-catalyst, $\text{Rh}(\text{CO})_2(\text{BINAP})\text{BF}_4$, which steers the carbofluorination transformation via allylic substitution/fluorination sequence, sidestepping the Heck-type allylation and cycloaddition reported by the authors' group. This protocol accommodates a wide range of alkenes, even tri-substituted ones, yielding mono-fluorides and gem-difluorides with 100% atom-economy under mild and simple conditions. The synthetic practicality was also demonstrated by gram-scale experiments and product derivatizations. Mechanistic experiments and DFT calculation rationalized the intriguing out-sphere mechanism of alkenes in allylic substitution and also how the C-F bond is formed. Overall, I fully recommend this work publication on Nature Communications, considering the following a few minor remarks addressed by the authors."

Thanks for the reviewer's positive comments and constructive suggestions, and we have addressed the following questions to improve our manuscript.

(b) Original comments: "It is reasonable that the stereocenter on the α -position (the position is regarding to the substrate aryl olefin) cannot be controlled as the C-F bond forms via a fluorine transfer to a carbon cation intermediate. However, as the C-C bond forms via outer-sphere allylic attack in which the Rh/Ligand is involved, readers would be interested in whether the stereocenter on the β -position can be controlled, in other words, how about the enantioselectivity of 1,2-disubstituted olefins by utilizing chiral ligands despite the low diastereoselectivity."

That's a great question! Actually, we have previously attempted to examine the enantioselectivity of the stereocenter on the β -position using β -methylstyrene as the

substrate. However, even after using all chiral HPLC columns available in our lab, the product, which includes four stereoisomers, still cannot reach baseline separation. Ulteriorly, we selected prop-1-ene-1,1-diyldibenzene as the substrate to avoid interference of diastereoisomers for testing enantioselectivity. Despite screening some BINAP derivatives, we have not yet found an appropriate ligand that can effectively control enantioselectivity on the β -position. R-BINAP gave the desired product with 6% ee, while R-BINAP^{3,5-Me₂} led to the best result (29% ee) at this present stage. These experimental results have been added in the revised supplementary information.

(c) Original comments: “Apart from the 1,2-disubstituted E-alkenes, how about the reactivity of Z-alkenes? The impact of the configuration of 1,2-disubstituted alkenes, including the reactivity and selectivity, warrants further investigation, which may also help understand the mechanism.”

That’s a great question! We investigated the effect of the configuration of 1,2-disubstituted alkenes at different ratio of *E/Z* under otherwise the same conditions. (see below). When using 1:1 *E/Z* mixture of alkene **2m**, the yield of **3m** was decreased to 75% with 1.5:1 dr. Treating the pure *Z*-**2m** under standard conditions resulted in a 50% yield of product **3m** with a 2.0:1 dr and incomplete conversion. Prolonging the reaction time to 48 hours increased the yield of **3m** to 66% with a comparable dr value. These results have been included in the revised supplementary information, which was briefly mentioned in the revised manuscript.

^aYield was determined by ¹H NMR using the 1,1,2,2-tetrachloroethane as the internal standard.

^bIsolated yield

(d) Original comments: "In comparison to the pre-mixed catalyst solution, it is interesting that the reactivity of this well-defined catalyst, $\text{Rh}(\text{CO})_2(\text{BINAP})\text{BF}_4$, is dramatically different in this reaction. When synthesizing this type of complex for research purposes, authors should take into account factors such as stability, storage conditions, and operational environment. This essential information is suggested to be provided to facilitate prompt utilization by readers who are interested in."

That's a great suggestion! While investigating this work, this series of Rh-catalysts was consistently kept outside. So far, there have been no irregularities in its catalytic activity and structure, indicating its stability in an air atmosphere. Furthermore, we have included additional relevant information and operational specifics regarding the synthesis procedure of this Rh-complex in the revised supplementary information.

(e) Original comments: "There is a recent transformation of gem-difluorinated cyclopropanes suggested to be cited, S. Qi, Y. Hua, L. Pan, J. Yang, J. Zhang, *Chin. J. Chem.* 2024, 42, 823-828."

Thank the reviewer to point out it. This reference was added as ref 36.

REVIEWERS' COMMENTS

Reviewer #1 (Remarks to the Author):

Since the authors have effectively addressed the concerns raised. The paper is ready for publication in its current state.

Reviewer #2 (Remarks to the Author):

The authors did a thorough and outstanding job addressing all of this reviewer's concerns. I can now recommend acceptance of the revised manuscript for publication in Nature Communications.